# The Association between *BZIP* Transcription Factors and Flower Development in *Litsea cubeba*

**DOI:** 10.3390/ijms242316646

**Published:** 2023-11-23

**Authors:** Siqi Wang, Yunxiao Zhao, Yicun Chen, Ming Gao, Yangdong Wang

**Affiliations:** 1State Key Laboratory of Tree Genetics and Breeding, Chinese Academy of Forestry, Beijing 100000, China; wsq980810@163.com (S.W.); zyx_yunxiao@163.com (Y.Z.); yicun_chen@163.com (Y.C.); 2Research Institute of Subtropical Forestry, Chinese Academy of Forestry, Hangzhou 310000, China

**Keywords:** *Litsea cubeba* gene family, flower development, degeneration of sexual organs, stage-specific differential expression, genome-wide identification

## Abstract

The basic leucine zipper (bZIP) family is one of the largest families of transcription factors among eukaryotic organisms. Members of the *bZIP* family play various roles in regulating the intricate process of flower development in plants. *Litsea cubeba* (Lour.) (family: Lauraceae) is an aromatic, dioecious plant used in China for a wide range of applications. However, no study to date has undertaken a comprehensive analysis of the *bZIP* gene family in *L. cubeba*. In this work, we identified 68 members of the *bZIP* gene family in *L. cubeba* and classified them into 12 subfamilies based on previous studies on *Arabidopsis thaliana*. Transcriptome data analysis revealed that multiple *LcbZIP* genes exhibit significantly high expression levels in the flowers of *L. cubeba*, while some also demonstrate distinct temporal specificity during *L. cubeba* flower development. In particular, some *LcbZIP* genes displayed specific and high expression levels during the stamen and pistil degradation process. Using differential gene expression analysis, weighted gene co-expression network analysis, and Gene Ontology enrichment analysis, we identified six candidate *LcbZIP* genes that potentially regulate stamen or pistil degradation during flower development. In summary, our findings provide a framework for future functional analysis of the *LcbZIP* gene family in *L. cubeba* and offer novel insights for investigating the mechanism underlying pistil and stamen degeneration in this plant.

## 1. Introduction

The basic leucine zipper (bZIP) family, one of the largest families of transcription factors in eukaryotic organisms, is characterized by the presence of a basic DNA binding region and an adjacent leucine zipper that facilitates dimerization [1]. The highly conserved basic region, with an invariant N-x7-R/K-x9 motif, is composed of approximately 16 amino acid residues [2]. In contrast, the Leu zipper region displays a lower degree of conservation and consists of multiple heptad repeats containing Leu or other hydrophobic amino acids with bulky side chains. The N-terminus of the basic region binds the major groove of double-stranded DNA, while the C-terminus of the Leu zipper forms a stacked coiled-coil structure, facilitating the homo and/or heterodimerization of bZIP proteins [3]. The *bZIP* family has multifaceted functions in plants, including enhancing plant resilience to biological and abiotic stress, coordinating diverse signaling pathways related to light, and regulating developmental processes [4,5,6]. For example, *OsbZIP23* acts as a transcriptional regulator in rice, controlling the expression of stress-related genes under abiotic stress conditions through an ABA-dependent regulatory pathway [7]. Additionally, *HY5* can transcriptionally regulate plant autophagy in response to light-to-dark conversion by interacting with *HDA9* [8]. Meanwhile, *GBF1* binds to the growth hormone response element and regulates the expression of growth hormone-responsive genes [9].

Notably, the *bZIP* gene family plays crucial roles in flower development, including the regulation of floral organ differentiation, pollen development, and embryonic development. For instance, *OsFD1* functions within complexes that promote flowering at the meristem, while *OsFD4*, which acts downstream of *OsFD1*, serves as a component of a florigen activation complex (FAC) that facilitates flowering at the shoot apical meristem [10]. *BZIP34* plays multiple roles in male reproductive development and has been linked to pollen germination, pollen tube growth, and exine patterning in sporophytes. Furthermore, *BZIP34* may function via sporophytic and/or gametophytic mechanisms in several metabolic pathways, including those associated with lipid metabolism regulation and cellular transport [11]. *AtbZIP18* has been indicated to have a potential inhibitory role in pollen and exhibits functional redundancy with *AtbZIP34* [12]. *Arabidopsis tga9* and *tga10*, encoding two bZIP transcription factors, interact with the floral glutaredoxins *roxy1* and *roxy2* and play redundant roles in anther development. Furthermore, it has been found that several members of the *bZIP* gene family regulate flower color formation and development [13]. Flower color, which refers to petals and sepals, is crucial for attracting pollinators, as well as for the identification of plant species. For example, the overexpression of *PgbZIP16* in Arabidopsis leads to a significant upregulation of the expression of the *UF3GT*, *ANS*, and *DFR* genes, resulting in increased anthocyanin accumulation and a subsequent alteration of flower color [14]. Members of the *bZIP* gene family also participate in the regulation of genes involved in flower development. For instance, the soybean gene *GmFDL19*, encoding a bZIP transcription factor, interacts with and is upregulated by *GmFT2a* and *GmFT5a*, two genes with redundant and differential roles in the regulation of flowering [15].

Litsea cubeba (Lour.), commonly known as mountain pepper, is a dioecious plant in the Lauraceae family. And it is an important aromatic plant in China with a wide range of applications. It can be used as a spice, a traditional Chinese medicine, and even as a source of raw materials for the dye or food additives industry [16,17]. During the early development of *L. cubeba* flowers, both pistil and stamen primordia are present. In female flowers, the pistil primordium develops normally, yielding a fertile pistil, while the stamen primordium develops abnormally, resulting in a degenerate and sterile stamen that lacks anther locules. In male flowers, stamens develop normally to form fertile stamens, whereas pistils develop abnormally, resulting in degenerate pistils. These degenerate pistils are characterized by smaller ovaries, the absence of stigma, and shortened or missing styles, which renders them sterile [18].

It is well documented that members of the *bZIP* gene family are extensively involved in flower development in various plants, particularly the development of stamens. However, it remains unclear whether members of this gene family are involved in regulating flower development in *L. cubeba*, especially in stamen degeneration.

In this work, we sought to identify and analyze members of the *bZIP* gene family in *L. cubeba*, a plant species in which the development of male and female flowers differs. Based on previous studies in *A. thaliana*, a total of 68 members of the *bZIP* gene family were identified and classified into 12 subfamilies. The physicochemical properties, gene structure, motif composition, chromosome distribution, and collinearity relationship of these genes were analyzed, and a comparative diagram was constructed for *L. cubeba* and other dicotyledonous and monocotyledonous plants. The expression patterns of these genes across different tissues and during the development of male and female flowers were also investigated based on transcriptome data. Meanwhile, six candidate genes with potential roles in pistil and stamen degeneration in *L. cubeba* were identified, and a weighted gene co-expression network analysis (WGCNA) and a Gene Ontology (GO) term enrichment analysis were conducted on these six genes. The findings of this study serve as a valuable resource for investigating the expression patterns of *bZIP* gene family members during flower development in *L. cubeba*. Furthermore, they establish a solid groundwork for an in-depth understanding of the mechanism underlying the degeneration of male and female flowers in this important plant species.

## 2. Results

### 2.1. Identification of LcbZIPs

In this study, we identified 68 proteins in the *bZIP* family using HMMER analysis (E-value < 1 × 10^−5^) [19,20]. The results were validated using the Pfam (PF00170) and SMART databases, which confirmed that the 68 *L. cubeba* proteins shared the bZIP domain, consistent with our predictions [20,21]. Based on their chromosomal locations, the validated genes encoding these proteins were named *LcbZIP1–LcbZIP68*.

We further analyzed the physical and chemical properties of the amino acid sequences of *L. cubeba* bZIP gene family members. The results showed that the proteins encoded via the 68 *LcbZIP* genes had molecular masses ranging from 11.14 (*LcbZIP28*) to 105.755.30 (*LcbZIP7*) kDa and pIs ranging from 4.62 (*LcbZIP4*) to 11.53 (*LcbZIP10*). Additionally, we found that the identified *bZIP* genes encoded proteins with lengths ranging from 98 (*LcbZIP28*) to 952 (*LcbZIP7*) amino acids. The GRAVY indexes ranged from 16.81–1.23 (*LcbZIP17*) to 0.74 (*LcbZIP60*), with 67 bZIP proteins demonstrating hydrophilic properties; they had negative GRAVY values. More detailed information is shown in Appendix A.

### 2.2. Phylogenetic Tree and Sequence Structure Analysis

To examine the evolutionary relationships and classification of the *bZIP* family members, we constructed an unrooted phylogenetic tree based on the amino acid sequence similarities between 68 *LcbZIPs* and 127 *AtbZIPs*, as previously documented [9] (Figure 1). Using a dendrogram based on previous studies in *A. thaliana*, we classified the 68 bZIP proteins into 12 groups [9,22]. Notably, no homologs of the *Arabidopsis* M subfamily were found among the LcbZIP proteins. The 12 groups differed in size, with the K and B groups being the smallest, each with only 1 member, and the A group being the largest, containing 15 members.

In this study, we used the protein sequences and annotation files of *LcbZIP* members to determine gene structure information (Figure 2). Our results showed that 11 of the 68 *LcbZIP* genes did not contain introns. The number of exons among the *LcbZIP* gene family members ranged from 1 to 19. Although the lengths of the introns varied, most homologs within the same subgroup shared similar exon/intron structures and numbers. For instance, all members in group H harbored six exons and three introns, while all members in group F contained three exons and one intron. Except for *LcbZIP10*, all the members of the S subfamily contained 1–2 exons and no introns.

Conserved motifs may participate in activating the function of *LcbZIP* proteins or indicate potential functional sites. To identify conserved motifs in the LcbZIP proteins, we used MEME and TBtools software v2.019 [22]. Our analysis identified 29 putative motifs among the *LcbZIPs*, with the number of motifs ranging from 1 to 9 among family members. Notably, closely related LcbZIP proteins in the same clade shared the same motif constitutions, which further confirmed the grouping results. For instance, all members of the S subfamily contained motifs 1 and 4, while all members of the D subfamily contained motifs 1, 2, 3, 5, 6, 11, and 12. Motifs 2, 3, 5, 6, and 12 were unique to D subfamily members, suggesting that they may have specific functions. Motif 1 was conserved in almost all *LcbZIPs* except for LcbZIP28, whereas motif 14 was specific to subfamilies B and G. Additionally, motifs 7, 8, and 9 were found only in subfamily A (Figure 2).

### 2.3. Chromosomal Locations and Collinearity Analysis of the LcbZIP Genes

The identification of the chromosomal locations of members of a gene family helps to reveal the origin and evolutionary processes of that family. In this study, we visualized the chromosomal distribution of the *LcbZIP* genes using genome annotation information and TBtools software [23] (Figure 3). Our results showed that the 68 *LcbZIP* genes were unevenly distributed on 12 chromosomes, with chromosome 5 containing the most *bZIP* genes (15), followed by chromosome 4 (9 genes). Chromosomes 10 and 11 had the fewest *bZIP* genes, with only one each. Notably, the number of *LcbZIP* genes distributed in each chromosome was not related to the length of the chromosome. For instance, chromosome 1, which is the largest chromosome, contained only 7 *LcbZIP* genes, while chromosome 7, which is substantially shorter than chromosome 1, contained 8.

Using TBtools and the MCScanX method to analyze segmental duplication events among the *LcbZIP* genes, we identified a total of 69 gene pairs that underwent segmental duplication [24]. Interestingly, these events were found to have occurred in 11 of the 12 chromosomes (Figure 4).

We also explored the collinearity relationships between the *bZIP* genes of *L. cubeba* and those of A. thaliana to identify orthologous genes and gain insights into the conservation of *bZIP* genes across different plant species. We found that the 68 *bZIP* genes of *L. cubeba* displayed collinearity relationships with 34 *bZIP* genes from *Arabidopsis* (Figure 5).

### 2.4. Tissue-Differential Expression of L. cubeba bZIP Genes

The inference of gene functions in plants can be facilitated by examining gene expression patterns in different tissues and organs. To investigate the tissue-specific expression patterns of the *LcbZIP* genes, we analyzed their expression in various tissues, including roots, stems, leaves, fruits, and flowers (Figure 6). The results were visualized using a heatmap, which revealed that most of the genes exhibited tissue-specific expressions. Eight genes (*LcbZIP45*, *LcbZIP15*, *LcbZIP63*, *LcbZIP24*, *LcbZIP27*, *LcbZIP42*, *LcbZIP66*, and *LcbZIP61*) displayed highly flower-specific expression patterns.

### 2.5. Expression Patterns of the LcbZIP Genes at Different Stages of Flower Development

The development of degenerate stamens/pistils in *L. cubeba* can be classified into the following three stages: primordia formation (M1: stamen primordium emergence, F1: pistil primordium appearance), stamen and pistil degeneration (M2: stamen degeneration, F2: pistil degeneration), and poststamen/pistil degeneration (M3: poststamen degeneration, F3: postpistil degeneration) [18]. To investigate the involvement of *LcbZIP* genes in *L. cubeba* flower development, we generated a heatmap using transcriptome data from these key developmental stages (Figure 7). Some *LcbZIP* genes, such as *LcbZIP34* and *LcbZIP40*, exhibited similar expression patterns across different stages of growth and development. This may suggest that these genes have a synergistic effect. Notably, the *LcbZIP* genes display temporal specificity. In male flowers, 10 genes (*LcbZIP8*, *LcbZIP24*, *LcbZIP18*, *LcbZIP20*, *LcbZIP31*, *LcbZIP3*, *LcbZIP4*, *LcbZIP49*, *LcbZIP15*, and *LcbZIP55*) showed specific high expression during the M2 period. In female flowers, 12 genes (*LcbZIP16*, *LcbZIP36*, *LcbZIP68*, *LcbZIP33*, *LcbZIP35*, *LcbZIP58*, *LcbZIP62*, *LcbZIP8*, *LcbZIP22*, *LcbZIP11*, *LcbZIP25*, and *LcbZIP27*) showed specific high expression during the F2 period. Additionally, 15 genes (*LcbZIP45*, *LcbZIP1*, *LcbZIP65*, *LcbZIP16*, *LcbZIP36*, *LcbZIP48*, *LcbZIP68*, *LcbZIP31*, *LcbZIP3*, *LcbZIP4*, *LcbZIP5*, *LcbZIP55*, *LcbZIP34*, *LcbZIP40*, and *LcbZIP39*) displayed highly distinct expression patterns between male and female flowers. These 30 genes are likely to be involved in sexual organ differentiation in *L. cubeba.*

### 2.6. Candidate Genes Involved in Sex Differentiation

Next, we identified the genes showing differential expression during key developmental stages using transcriptomic data. To construct a co-expression network, we employed weighted gene co-expression network analysis (WGCNA) on the DEGs identified in the RNA-Seq data [25]. Of the 30 aforementioned differentially expressed *LcbZIP* genes, 6 (*LcbZIP5*, *LcbZIP40*, *LcbZIP18*, *LcbZIP20*, *LcbZIP22*, and *LcbZIP35*) were found to exhibit strong co-expression relationships with other genes in this network. Additionally, we identified six co-expression networks, as illustrated in Figure 8.

The largest of these networks was centered on *LcbZIP35* and consisted of 990 genes, while the smallest network, centered on *LcbZIP40*, comprised only 208 genes. To gain a better understanding of the functional characteristics of these six candidate genes, we subsequently performed a GO term enrichment analysis (Figure 9). The co-expressed genes associated with *LcbZIP40* were found to be enriched in pathways related to floral organ development, flower development, and floral whorl development. Additionally, the genes co-expressed with *LcbZIP18* were enriched in pathways such as photoperiodism, flowering, and pollen tube growth. Similarly, the genes co-expressed with *LcbZIP20* were mostly involved in flowering-related pathways and the regulation of plant development, particularly heterochronic development processes. Furthermore, genes in the co-expression network centered around *LcbZIP22* were found to be enriched in microsporocyte differentiation and sporocyte differentiation pathways. Finally, the co-expressed genes centered around *LcbZIP5* and *LcbZIP35* play significant roles in multiple metabolic processes, cellular growth regulation, and nucleic acid processing.

### 2.7. Validation of the DEGs via qRT–PCR

To validate the results obtained from the RNA-Seq analysis, we measured the expression levels of six genes identified as being differentially expressed during key developmental stages (Figure 10). The expression patterns of only two genes, *LcbZIP5* and *LcbZIP40*, showed significant differences throughout male and female flower development. *LcbZIP18* and *LcbZIP20* showed specific and high expression during the M2 stage in male flowers. Conversely, *LcbZIP22* and *LcbZIP35* displayed specific and high expression during the F2 stage in female flowers. We generated a heatmap using the expression data of these six genes. As shown in Figure 10, the RNA-Seq and qRT–PCR results were consistent.

## 3. Discussion

bZIP proteins, comprising an important class of transcription factors, have been identified in many plant species, including *pomegranate*, *Cannabis sativa L.*, *poplar*, *olive*, *castor bean*, *Oryza sativa*, *Banana*, and *Vitis vinifera* [3,14,26,27,28,29,30]. However, no similar study has been conducted in *L. cubeba*. In this work, we identified 68 *LcbZIP* genes and classified them into 12 subfamilies based on the relationships between the *bZIP* genes of *L. cubeba* and those of *A. thaliana*. This classification method helps to better understand the evolutionary history and functional diversity of the *LcbZIP* gene family. In *Arabidopsis*, subfamily-specific and conserved motifs can contribute significantly to the functional differentiation of *AtbZIP* subfamilies. For example, some members of the D group are involved in flower development, such as *TGA8/PAN* (PERIANTHIA), which controls the formation of floral organ primordia [31]. The *LcbZIP* genes within group D could have similar functions. The genomic structure of each gene, particularly the number and distribution of exons/introns, can serve as a crucial record of key evolutionary events and provide insights into understanding the emergence and evolution of a given gene. In this study, we found that genes within the same subfamily have similar intron/exon structures and conserved domains. This conservation in structure and sequence may be related to functional similarity and the conservation of regulatory mechanisms. Furthermore, we investigated the chromosomal distribution of the *LcbZIP* genes and found that their genomic distribution was not uniform. This is consistent with studies conducted on poplar and rose [29,32]. The uneven distribution of the *LcbZIP* genes among chromosomes could be attributed to factors such as gene duplication, translocation, and deletion events that occurred during the evolution of *L. cubeba* [26]. Additionally, this uneven distribution may have been influenced by functional divergence and selection pressures [33]. We also observed a certain degree of clustering of the *bZIP* genes on the chromosomes, which may be closely related to the functions of the gene family, as the proximity of genes on a chromosome may contribute to their interaction and coordinated expression in regulatory networks. This chromosomal distribution pattern may also be associated with the origin and evolution of the gene family and provides clues for further study of the functional and regulatory mechanisms underlying the roles of *LcbZIP* family members. Collinearity refers to the phenomenon in which the order of two or more genes on a chromosome is highly conserved. Collinearity analysis plays an important role in understanding the structure and evolution of genomes. In this study, we analyzed the collinearity within the *L. cubeba* species and identified 69 collinear gene pairs. This strongly suggests that segmental duplication events played a significant role in driving *LcbZIPs* diversity. These genes may have similar functions or be involved in related regulatory relationships. The existence of collinearity may be associated with gene family amplification and duplication events [34]. Interspecies collinearity analysis revealed that a total of 68 *LcbZIP* genes exhibited collinearity relationships with 34 *Arabidopsis* genes, suggesting that these *bZIP* genes have maintained orthologous relationships in these two species.

BZIP transcription factors play a significant role in plant development and growth by enhancing plant resistance to biotic and abiotic stress and coordinating diverse signaling pathways associated with light, environmental stimuli, and developmental processes [35]. In particular, members of the *bZIP* family have been reported to participate in plant flower development through several pathways. For example, in *tobacco* (*Nicotiana tabacum*), the bZIP transcription factor *BZI-4* can interact with *BZI-1*, thereby affecting the size of the floral organs. *BZI-1* is involved in the regulation of carbohydrate supply during pollen development, and the presence of its dominant–negative form, *BZI-1 δ N*, results in severe defects in this process [36]. Previous studies have demonstrated that the bZIP transcription factor *FD* can interact with *FLOWERING LOCUS T* (*FT*), thereby regulating flowering. In addition, *FD* has been found to interact with *TERMINAL FLOWER 1* (*TFL1*), which is a floral repressor. Moreover, the bZIP transcription factor *AREB3*, which is associated with *FD,* is expressed in a spatiotemporal pattern in the shoot apical meristem. Its expression strongly overlaps with that of *FD* and is involved in the transmission of *FT* signaling [37,38].

To investigate the expression patterns of *LcbZIP* genes across different tissues, we generated a heatmap. Most genes showed significant tissue specificity. For example, we observed that *LcbZIP45*, *LcbZIP15*, *LcbZIP66*, and *LcbZIP24* exhibited significant changes in expression during flower development. This suggests that these genes may play important roles in the development and maturation of *L. cubeba* flowers. Analysis of the heatmap of *LcbZIP* genes during different stages of flower development indicated that their expression exhibits significant temporal specificity. We identified genes that were specifically highly expressed during the stamen degradation and carpel degradation stages, as well as genes that showed differential expression during the development of male and female flowers. These genes were then subjected to WGCNA, leading to the identification of six candidate genes (*LcbZIP5*, *LcbZIP40*, *LcbZIP18*, *LcbZIP20*, *LcbZIP22*, and *LcbZIP35*) that are likely to be involved in regulating stamen and carpel degradation. These stage-specific expression patterns may be related to pistil and stamen degeneration in *L. cubeba*. To understand their potential functions, we mapped these genes to the *Arabidopsis* genome. The best match for *LcbZIP40* was *AT5G06839*, which encodes the *TGA10* transcription factor. *AT1G08320* is the *LcbZIP5* counterpart and codes for the *TGA9* transcription factor. The activation of *TGA9/10* through the ROXY-mediated modification of cysteine residues is important for anther development in *Arabidopsis* [13]. These observations suggest that *LcbZIP40* and *LcbZIP5* may play crucial roles in anther development in *L. cubeba*. *AT4G35900*, which can interact with FT and activate the expression of *APETALA1*, was the best match for *LcbZIP22* [39]. *AT1G49720* (*ABF1*) is the counterpart for *LcbZIP18*. *ABF1* interacts with *FIE2*, leading to the recruitment of the *PRC2* complex, which deposits the suppressive *H3K27me3* histone modification on *Ehd1* and *Ehd2*, resulting in the inhibition of their transcription and, ultimately, delayed flowering [1]. Taken together, the DEGs specific to these stages are likely to have significant roles in flower development in *L. cubeba*.

In the GO enrichment analysis, the genes coexpressed with *LcbZIP40* were found to be enriched in pathways related to floral organ development, flower development, and floral whorl development. This suggested that *LcbZIP40* may play an important role in the development of floral organs and floral whorls, as well as overall flower development. Additionally, the *LcbZIP40* gene was found to be enriched in pathways associated with plant hormones, such as those involved in the maintenance of brassinosteroid homeostasis and the gibberellic acid-mediated signaling pathway. This implied that *LcbZIP40* may influence flower development by regulating plant hormone synthesis, metabolism, and signal transduction. The genes coexpressed with *LcbZIP18* were observed to be mainly involved in pathways such as photoperiodism, flowering, and pollen tube growth, suggesting that *LcbZIP18* may be involved in the regulation of photoperiodism and flowering time in plants and may also be important for the growth and guidance of pollen tubes. The genes coexpressed with *LcbZIP20* were enriched in the regulation of plant development, particularly heterochronic development processes. This suggested that *LcbZIP20* may have a regulatory function in the determination of the timing of plant development. *LcbZIP20* is also involved in photoperiodism and flowering-related pathways, indicating that it may have a crucial function in plant perception and photoperiod regulation, specifically, the control of flowering time. In conclusion, *LcbZIP20* may exert significant regulatory control over various aspects of plant flower development, including heterochronic development, inflorescence formation, and the regulation of flowering time. Genes in the coexpression network centered around *LcbZIP22* were found to be enriched in microsporocyte and sporocyte differentiation pathways. This suggested that *LcbZIP22* may regulate the differentiation of pollen mother cells, thus making a key contribution to pollen formation. The results of the enrichment analysis further showed that *LcbZIP22* is likely to be involved in multiple hormone signaling pathways, including those associated with auxin signaling. This observation suggested that *LcbZIP22* may regulate flower development through hormone signaling pathways. GO enrichment analysis involving the coexpressed genes centered on *LcbZIP5* and *LcbZIP35* indicated that *LcbZIP5* plays an important role in multiple metabolic processes, cellular growth regulation, and nucleic acid processing. It may also be involved in regulating the growth and metabolic activity of flower cells, as well as the regulation of gene expression. *LcbZIP35* may be involved in regulating chloroplast RNA processing and function, protein transport and nuclear localization, RNA modification and processing, and the regulation of metabolic pathways.

## 4. Materials and Methods

### 4.1. Plant Materials

The *L. cubeba* materials were obtained from Hang Zhou City, Zhejiang Province, China (30°27′94″ N, 119°58′43″ E). Roots, stems, leaves, flowers, and fruits were collected for analysis. Quantitative gene expression values obtained from plant materials at different stages of flower bud differentiation were used to generate heatmaps. Based on the findings of Xu Zilong, female flower buds 1.2 mm in diameter were chosen for the F1 stage, while flower buds measuring 1.7 and 2.6 mm in diameter were selected for the F2 and F3 stages, respectively [18]. For male flower buds, those with a diameter of 1.4 mm were collected for the M1 stage, while buds measuring 1.8 and 2.9 mm in diameter were selected for the M2 and M3 stages, respectively.

### 4.2. Identification of LcbZIP Genes

We retrieved the coding sequences and protein sequences essential for analysis from the *L. cubeba* genome database. [40]. TBtools v2.019 (https://github.com/CJ-Chen/TBtools, accessed on 10 May 2023) was used to extract the *L. cubeba bZIP* gene sequences with the screening thresholds set to an E-value < 1 × 10^−5^ and ≥50% identity match HMMER, and BLAST results were analyzed to identify putative LcbZIP proteins [41,42]. Any duplicated sequences were manually removed. The obtained candidate genes were then further verified using the Pfam and SMART databases. In total, 68 *LcbZIP* transcription factors were identified. A phylogenetic tree was constructed with PhyML 3.0 using default parameters [43].

### 4.3. Analysis of bZIP Protein Sequences and Gene Structure

The LcbZIP protein sequences were submitted to the ExPASy online program for calculating the molecular masses, isoelectric points (pI), and grand average of hydropathicity (GRAVY) indexes of the putative proteins [44,45]. MEME was used to identify conserved motifs within the *bZIP* gene family [22,46]. The resulting file was subsequently imported into TBtools for visualization. The motif annotation information was sourced from the SMART and Pfam databases [47,48]. The Gff3 files of the *LcbZIP* genes were submitted to TBtools for gene structure analysis [23].

### 4.4. Chromosomal Location and Collinearity Analysis of the LcbZIP Genes

BLAST programs were utilized to map the *LcbZIP* sequences onto the *L. cubeba* chromosome survey sequences to identify the locations of the *LcbZIP* genes on the 12 chromosomes. MG2C v2.1 software (http://mg2c.iask.in/mg2c_v2.1/, accessed on 6 June 2023) was used to display the exact gene locations. Furthermore, to examine interspecies collinearity among the *LcbZIP* genes, we conducted an analysis using MCScanX in TBtools software v2.019. (https://github.com/CJ-Chen/TBtools, accessed on 6 June 2023) [49].

### 4.5. Extraction of RNA and Quantitative Reverse Transcription-PCR

Total RNA was extracted from *L. cubeba* and reverse-transcribed using the method described by Zhao et al. [50]. qPCR was performed on an ABI PRISM 7500 instrument using TB Green Premix Ex Taq. The reference gene for L. cubeba established by Chen et al. was the ubiquitin-conjugating enzyme (UBC) gene. Relative expression levels were calculated using the 2^−ΔΔCT^ method. The results are presented as the means ± standard deviations of three replicates. The qPCR primer sequences were designed using Primer Premier 3.0 software (https://primer3.ut.ee/, accessed on 6 June 2023) and are listed in Appendix A.

### 4.6. Gene Coexpression Networks and GO Analysis

Gene coexpression networks were identified using the WGCNA package in R [25]. Differential gene expression analysis was performed on the transcriptome data for *L. cubeba* flower buds to identify key stage-specific differentially expressed genes (DEGs). Subsequently, a gene coexpression network was constructed using the data of these DEGs [51]. The results were visualized in Cytoscape [52]. GO term enrichment analysis of the six key genes was conducted using OmicShare Tools (https://www.oicshare.com/tools/Home/Soft/getsoft, accessed on 6 June 2023), focusing on the biological processes category [51]. The Benjamini–Hochberg method was used to calculate and adjust the *p* values for each pathway [53].

## 5. Conclusions

In conclusion, we identified 68 *bZIP* gene family members in *L. cubeba* and focused on investigating the function of members of this family in flower development, particularly their role in the degeneration of male and female reproductive organs. We further analyzed the physicochemical properties, gene structure, chromosomal distribution, collinearity relationships, and gene expression patterns of the *LcbZIP* genes in different tissues and at different stages of flower development. Additionally, we identified stage-specific gene expression patterns and ultimately identified six *LcbZIP* genes—*LcbZIP40*, *LcbZIP5*, *LcbZIP35*, *LcbZIP20*, *LcbZIP22*, and *LcbZIP18*—as being potentially involved in the degeneration of male and female reproductive organs. Based on the identification of these six key regulators and their associated gene networks, we discovered additional regulators and their corresponding gene networks involved in diverse biological processes.

## Figures and Tables

**Figure 1 ijms-24-16646-f001:**
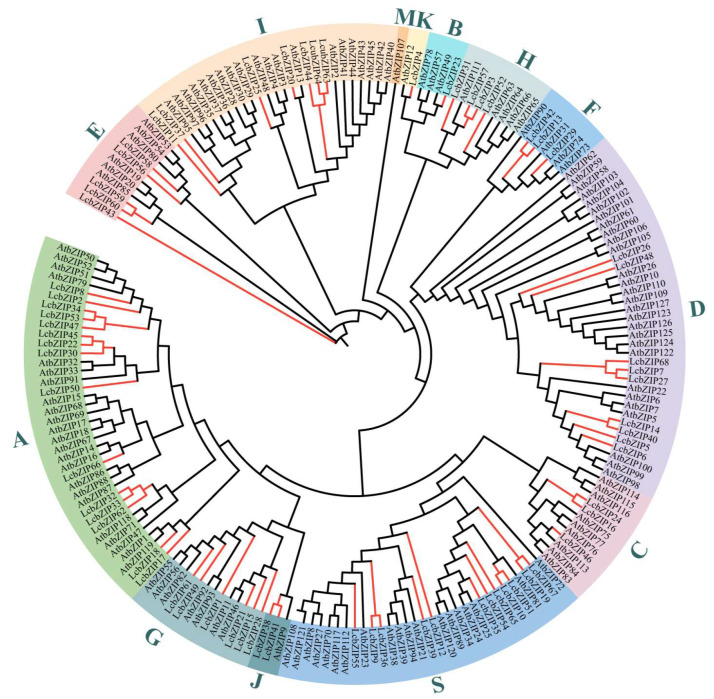
A dendrogram of *L. cubeba* and *Arabidopsis bZIP* members. The dendrogram was drawn in TBtools. Different colors were assigned to distinguish the different groups. These groups were labeled using letters representing key members (A for ABF/AREB/ABI5, C for CPRF2-like, G for GBF, H for HY5), protein size (B for big, S for small), or in alphabetical order.

**Figure 2 ijms-24-16646-f002:**
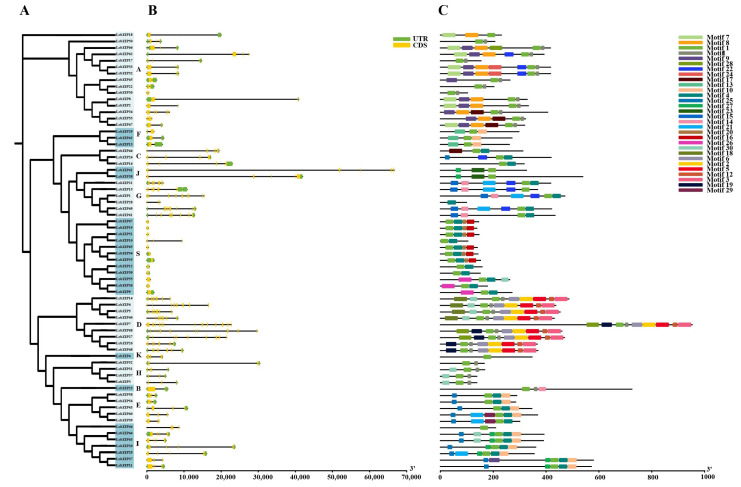
DNA structure and protein motifs of the *bZIP* gene family in *Litsea cubeba*. (**A**) Phylogenetic tree for each subfamily. Different letters represent different subfamilies. (**B**) Gene structure. Exons and the 3′UTRs/5′UTRs are displayed using black lines and yellow bars, respectively. Black dotted lines denote introns. (**C**) Protein motifs in the bZIP members. The colored boxes represent different motifs. Clustering was performed according to the results of the phylogenetic analysis.

**Figure 3 ijms-24-16646-f003:**
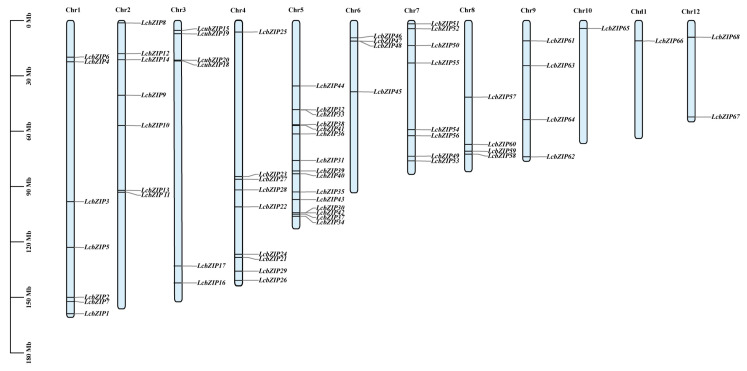
Chromosomal distribution of the *Litsea cubeba bZIP* genes. Chr1–12 represent chromosomes 1–12.

**Figure 4 ijms-24-16646-f004:**
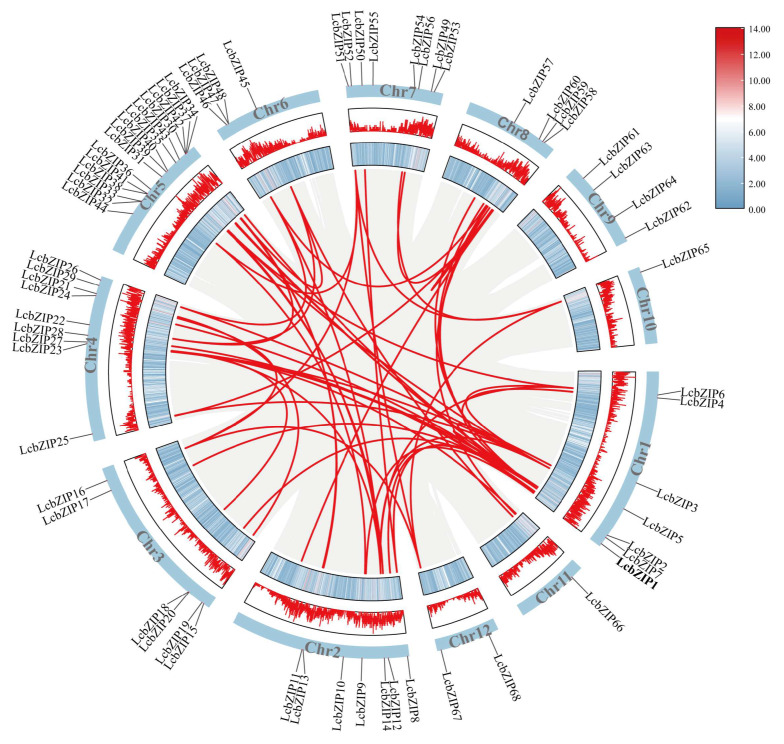
Collinearity analysis of the bZIP gene family in *Litsea cubeba*. Chromosomes 1–12 are represented by yellow rectangles. The lines, heatmaps, and histograms along the rectangles represent gene density on the chromosomes. The gray lines indicate synteny blocks in the *L. cubeba* genome, while the red lines between chromosomes indicate gene pairs arising from segmental duplication.

**Figure 5 ijms-24-16646-f005:**
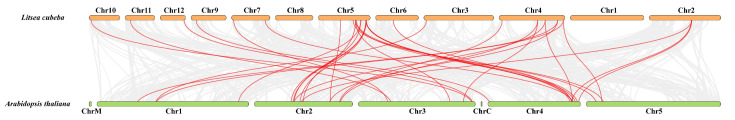
Synteny analysis of the *bZIP* genes between *Litsea cubeba* and *Arabidopsis thaliana*. The gray lines indicate gene blocks in *L. cubeba* that are orthologous to gene blocks in *A. thaliana*. The red lines indicate syntenic *LcbZIP* gene pairs. The orange bar represents the chromosomes of the *L. cubeba.* The green bars represent the chromosomes of *Arabidopsis thaliana*.

**Figure 6 ijms-24-16646-f006:**
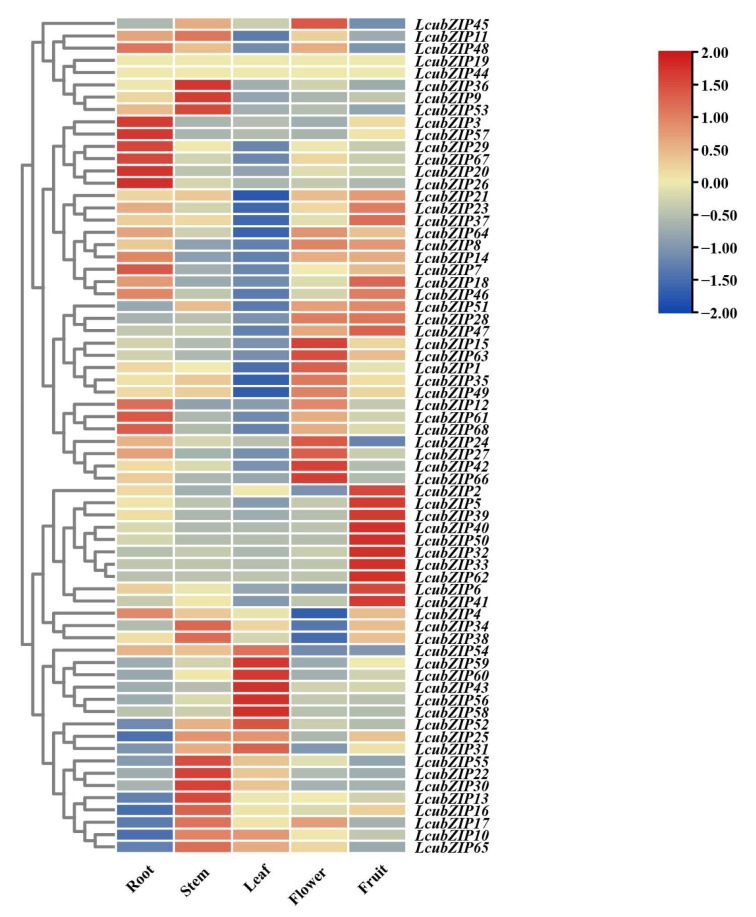
Heatmap of the expression of the 68 *LcbZIP* genes in the root, stem, leaf, flower, and fruit.

**Figure 7 ijms-24-16646-f007:**
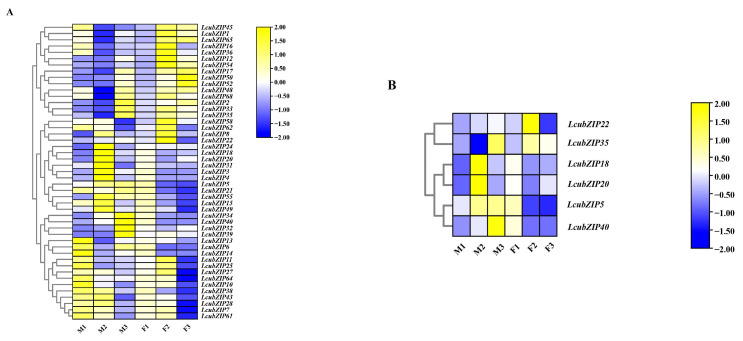
Heatmap of the expression of *LcbZIP* genes at different time points. M1: stage of stamen primordium emergence; M2: stamen degeneration stage; M3: post-stamen degeneration stage. F1: stage of pistil primordium appearance; F2: pistil degeneration stage; F3: post-pistil degeneration stage. (**A**): Heatmap of the expression of all *LcbZIP* genes at different time points. (**B**): Heatmap of the expression of six candidate *LcbZIP* genes at different time points.

**Figure 8 ijms-24-16646-f008:**
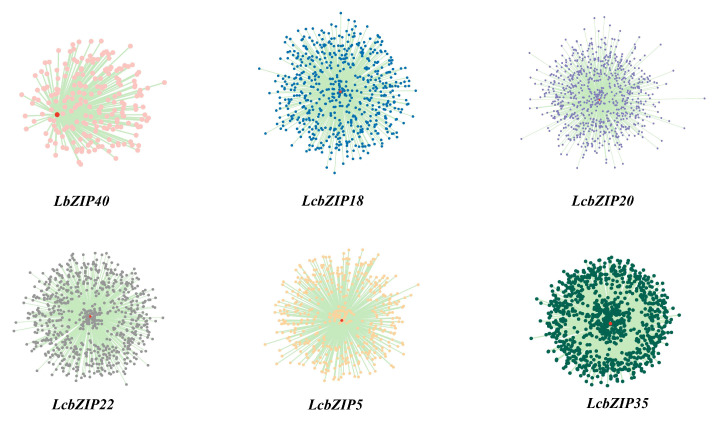
Transcription factor-centered co-expression network for six differentially expressed genes. The dots represent genes, and the lines indicate that the genes have co-expression relationships.

**Figure 9 ijms-24-16646-f009:**
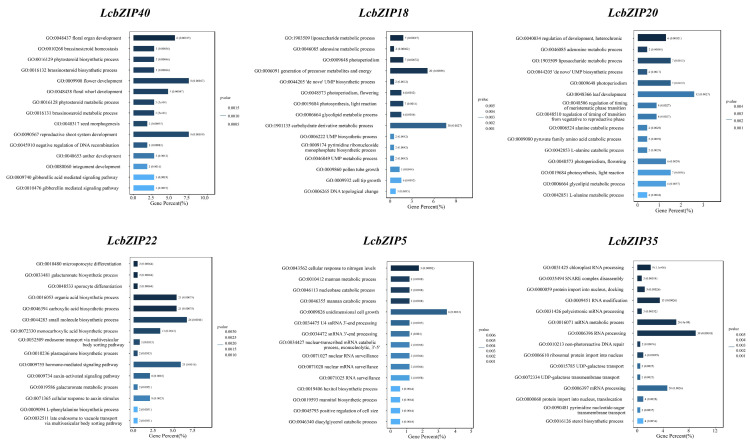
Gene Ontology (GO) enrichment analysis of the six sets of co-expressed genes.

**Figure 10 ijms-24-16646-f010:**
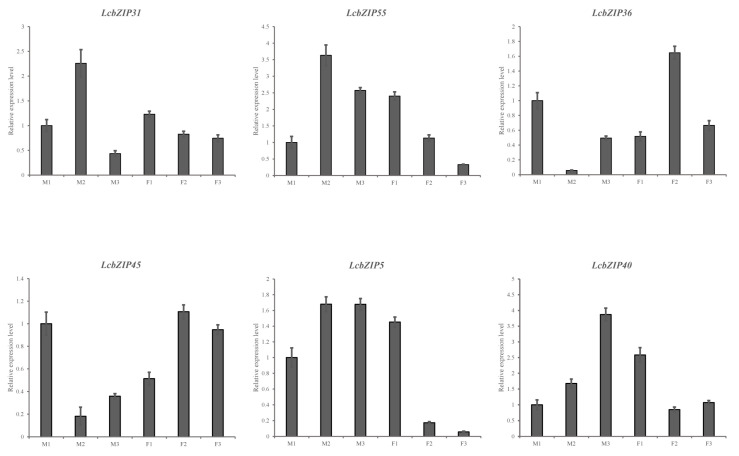
Gene expression levels based on RNA-Seq and qPCR. Data are presented as means ± standard deviation of three replicates. The *Litsea cubeba ubiquitin-conjugating enzyme* (*UBC*) gene served as an internal reference. The error bars are standard deviations from the biological replicates.

## Data Availability

The datasets supporting the conclusions of this article are available in the NCBI Short Read Archive under accession number PRJNA763042. https://www.ncbi.nlm.nih.gov/bioproject/PRJNA763042, accessed on 15 September 2022. And SRR10053824, SRR10053795, SRR10053793, SRR10053782, SRR10053780, SRR10053770, SRR10053769, SRR10053767, SRR10053765, SRR10053109, SRR10052556, SRR10052491, SRR10052460, SRR10052459, SRR10052050, SRR10052049, SRR10051549, SRR10051547.

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
