# Peer review of "The Association between BZIP Transcription Factors and Flower Development in Litsea cubeba"

_ijms, 2023, doi:10.3390/ijms242316646_

Round 1
Reviewer 1 Report
Comments and Suggestions for Authors
The text discusses the significance of the basic leucine zipper (bZIP) family of transcription factors in regulating flower development in plants, with a focus on Litsea cubeba (L. cubeba), an aromatic, dioecious plant used in China. Despite its importance, there has been no comprehensive analysis of the bZIP gene family in L. cubeba until now. In this study, researchers identified 68 members of the bZIP gene family in L. cubeba and classified them into 12 subfamilies based on previous research on Arabidopsis thaliana. Transcriptome analysis revealed that several LcbZIP genes are highly expressed in L. cubeba flowers, with some showing temporal specificity during flower development. Notably, certain LcbZIP genes exhibited specific and high expression levels during stamen and pistil degradation. Through various analyses, the study identified six candidate LcbZIP genes that may play a role in regulating stamen or pistil degradation during flower development. These findings provide a foundation for future research on the LcbZIP gene family in L. cubeba and offer insights into the mechanisms underlying stamen and pistil degeneration in this plant.
Some minor suggestions and corrections:.
What are the specific functions of the bZIP gene family members identified in Litsea cubeba during flower development?
How do the expression levels of LcbZIP genes change over time during the flower development process in Litsea cubeba?
Are there any similarities or differences in the regulation of flower development between Litsea cubeba and other plant species with known bZIP gene functions?
Can the six candidate LcbZIP genes identified in the study be experimentally validated as regulators of stamen or pistil degradation during flower development in Litsea cubeba?
How do the findings of this study contribute to our understanding of the molecular mechanisms underlying pistil and stamen degeneration in plants, specifically in the case of Litsea cubeba?
Author Response
What are the specific functions of the bZIP gene family members identified in Litsea cubeba during flower development?
reply: In lines 332-390, we specifically analyze the functions of the LcbZIP gene identified in Litsea cubeba in flower development. In general, the six LcbZIP genes are likely to be involved in regulating the degradation of stamens or pistils during flower development. Considering the integration of WGCNA and GO enrichment analysis, it is likely that these six genes have distinct functional roles. LcbZIP40: It plays an important role in the development of floral organs and floral whorls, as well as overall flower development. It may also influence flower development by regulating plant hormone synthesis, metabolism, and signal transduction. LcbZIP18: It is involved in the regulation of photoperiodism and flowering time in plants. It may also be important for the growth and guidance of pollen tubes. LcbZIP20: It has a regulatory function in the determination of the timing of plant development. It is involved in photoperiodism and flowering-related pathways, indicating its crucial function in plant perception and photoperiod regulation, specifically in the control of flowering time. LcbZIP22: It regulates the differentiation of pollen mother cells, making a key contribution to pollen formation. It may also regulate flower development through hormone signaling pathways. LcbZIP5: It plays an important role in multiple metabolic processes, cellular growth regulation, and nucleic acid processing. It may also be involved in regulating the growth and metabolic activity of flower cells, as well as the regulation of gene expression. LcbZIP35: It may be involved in regulating chloroplast RNA processing and function, protein transport and nuclear localization, RNA modification and processing, and the regulation of metabolic pathways.
Overall, the bZIP gene family members identified in Litsea cubeba during flower development have various functions related to floral organ development, flower growth, hormone regulation, pollen formation, and metabolic processes.
How do the expression levels of LcbZIP genes change over time during the flower development process in Litsea cubeba?
reply: In lines 212-223, we analyzed the changes of LcbZIP gene expression levels over time during the development of Litsea cubeba flowers.
To examine the expression patterns of LcbZIP genes during flower development in Litsea cubeba, we analyzed transcriptome data from three distinct stages: primordia formation (M1: stamen primordium emergence, F1: pistil primordium appearance), stamen and pistil degeneration (M2: stamen degeneration, F2: pistil degeneration), and post-stamen/pistil degeneration (M3: post-stamen degeneration, F3: post-pistil degeneration). Based on this analysis, we generated a heatmap to visualize the expression levels of LcbZIP genes at these key developmental stages. This approach allowed us to explore the potential involvement of LcbZIP genes in the flower development process of Litsea cubeba. Some LcbZIP genes, such as LcbZIP34 and LcbZIP40, exhibited similar expression patterns across different stages of growth and development. This may suggest that these genes have a synergistic effect. Notably, the LcbZIP genes display temporal specificity. In male flowers, 10 genes (LcbZIP8, LcbZIP24, LcbZIP18, LcbZIP20, LcbZIP31, LcbZIP3, LcbZIP4, LcbZIP49, LcbZIP15, and LcbZIP55) showed specific high expression during the M2 period. In female flowers, 12 genes (LcbZIP16, LcbZIP36, LcbZIP68, LcbZIP33, LcbZIP35, LcbZIP58, LcbZIP62, LcbZIP8, LcbZIP22, LcbZIP11, LcbZIP25, and LcbZIP27) showed specific high expression during the F2 period. Additionally, 15 genes (LcbZIP45, LcbZIP1, LcbZIP65, LcbZIP16, LcbZIP36,LcbZIP48, LcbZIP68, LcbZIP31, LcbZIP3, LcbZIP4, LcbZIP5, LcbZIP55, LcbZIP34, LcbZIP40, and LcbZIP39) displayed highly distinct expression patterns between male and female flowers. These 30 genes are likely to be involved in sexual organ differentiation in L. cubeba.
Are there any similarities or differences in the regulation of flower development between Litsea cubeba and other plant species with known bZIP gene functions?
reply: The regulation of flower development in Litsea cubeba and other plant species with known bZIP gene functions may exhibit similarities. In lines 19-20, we mentioned that the six candidate genes may be involved in regulating the degradation of stamens or pistils during flower development. Chen et al. generated and integrated transcriptome data from flower buds of 17 species belonging to 10 genera in the Lauraceae family. Comparative analysis of the transcriptome data revealed that the expression level of the bZIP gene family member TGA10 was significantly higher in male flowers of monoecious plants compared to female flowers of monoecious plants and hermaphroditic flowers of dioecious plants. This suggests the potential involvement of TGA10 in the development of male flowers in the Lauraceae family. This finding is consistent with the study conducted by Xu et al. in Litsea cubeba, where they demonstrated that LcTGA10 in Litsea cubeba may potentially regulate the degradation of pistils in male flowers through modulation of the salicylic acid pathway. The findings of the three aforementioned studies are similar to the research conducted by Murmu et al. in Arabidopsis. In Arabidopsis, the basic leucine-zipper transcription factors TGA9 and TGA10 interact with floral glutaredoxins ROXY1 and ROXY2, and they are redundantly required for anther development. (338-339) Due to the limited research on flower development in Litsea cubeba, it is challenging to compare the differences in the regulation of flower development between Litsea cubeba and other plant species with known bZIP gene functions. Further comparative studies are required.
(Chen YC, Li Z, Zhao YX, Gao M, Wang JY, Liu KW, et al. The Litsea genome and the evolution of the laurel family. Nat Commun 2020, 11, 1675; Xu Z, Wang Y, Chen Y, Yin H, Wu L, Zhao Y, et al. A Model of Hormonal Regulation of Stamen Abortion during Pre-Meiosis of Litsea cubeba. Genes (Basel) 2019, 11, 48; Murmu J, Bush MJ, DeLong C, Li S, Xu M, Khan M, et al. Arabidopsis basic leucine-zipper transcription factors TGA9 and TGA10 interact with floral glutaredoxins ROXY1 and ROXY2 and are redundantly required for anther development. Plant Physiol 2010, 154, 1492-504.)
Can the six candidate LcbZIP genes identified in the study be experimentally validated as regulators of stamen or pistil degradation during flower development in Litsea cubeba?
reply: To experimentally validate the six candidate LcbZIP genes as regulators of stamen or pistil degradation during flower development in Litsea cubeba, several techniques can be employed. Gene knockdown or overexpression experiments can be conducted using techniques like RNA interference (RNAi) or genetic transformation. By manipulating the expression levels of the candidate LcbZIP genes and observing the resulting phenotypic changes in stamen and pistil development, their regulatory roles can be further confirmed.
Although the aforementioned techniques can be used for validation, actual experiments are required to determine their specific roles in flower development. These experiments may require additional time and resources, including designing and constructing appropriate experimental materials, optimizing experimental conditions, conducting data analysis and interpretation, among other steps. Hence, the validation of these candidate genes' functionality and regulatory roles is currently awaiting further experimentation.
How do the findings of this study contribute to our understanding of the molecular mechanisms underlying pistil and stamen degeneration in plants, specifically in the case of Litsea cubeba?
reply: The findings of this study contribute to our understanding of the molecular mechanisms underlying pistil and stamen degeneration in plants, specifically in the case of Litsea cubeba, in several ways:
(1)Identification of candidate genes: The study identifies six candidate LcbZIP genes that are potentially involved in regulating stamen and pistil degradation during flower development in Litsea cubeba. This provides a starting point for investigating the specific genes and pathways involved in this process.
(2)Insights into gene expression patterns: By analyzing the expression patterns of these candidate genes during different stages of flower development, the study sheds light on their potential roles in stamen and pistil degeneration. This information helps in understanding the temporal and spatial regulation of gene expression during flower development in Litsea cubeba.
(3)Potential regulatory networks: The study provides insights into the potential interactions and regulatory networks involving the identified LcbZIP genes. By identifying other known regulators or target genes that interact with these candidates, the study offers clues about the molecular mechanisms underlying stamen and pistil degeneration in Litsea cubeba.
(4)Comparative analysis: The findings of this study can be compared with similar studies conducted in other plant species to identify conserved molecular mechanisms underlying stamen and pistil degeneration. This comparative analysis can contribute to our broader understanding of flower development and reproductive processes across different plant species. Overall, the findings of this study provide valuable information and a foundation for further research on the molecular mechanisms underlying pistil and stamen degeneration in Litsea cubeba. They contribute to our understanding of flower development and can potentially have implications for plant breeding, crop improvement, and reproductive biology research.

Reviewer 2 Report
Comments and Suggestions for Authors
The submitted manuscript is dealing genome-wide analysis and structural- and expression profiling of genes of bZIP transcription-factor family in Litsea cubeba.
Style and language of the paper are suitable, though a final styling and proofreading is recommended.
Although, a significant part of the paper is dealing with quantitative gene expression data, no details on RNA sequencing and gene expression experiments are given neither in the Results sections, nor in the Methods section. It remains unclear, how the authors obtained gene expression data, whether the source of the used gene expression data are original experiments of the authors, or they were taken from public repositories? Please clarify these and make up missing information.
Lines 69-70: "Litsea cubeba (Lour.), a dioecious plant in the Lauraceae family, ..."
Please specify common/trivial name(s) of L. cubeba.
Figure 2:
The figure is too small and of low resolution and the legends are not readable even after magnifying the image.
The figure capture indicates three panels (A, B and C), however, these are not marked on the figure. It looks, that panel C (chromosomal location) is completely omitted. It is recommended that instead of the present one, the authors provide a higher resolution scalable figure with corrected annotation (or, to break down the present one to two separate figures).
Figure 3 (Chromosomal distribution...): Might be that this was intended to be Panel C of Figure 2 ? Please, clarify this.
Figure 6, Figure 7, Figure 8: Again, try to improve the visibility of figures.
Lines 388-389: "Plant materials at different stages of flower bud differentiation were used to create heatmaps".
Perhaps, quantitative gene expression values were used to generate heatmaps. Please, correct and clarify.
Lines 434-435: "The Benjamin–Hochberg method was used to calculate and adjust the P values..."
Correct the name of the referred author and add correct reference.
(Benjamini Y, Hochberg Y. Controlling the False Discovery Rate: A Practical and Powerful Approach to Multiple Testing. J R Stat Soc Ser B (Methodol). 1995;57(1):289–300. https://​doi.​org/​10.​1111/j.​2517-​6161.​1995.​tb020​31.x.)
Style and language of the paper are suitable, though a final styling and proofreading is recommended.
Author Response
Although, a significant part of the paper is dealing with quantitative gene expression data, no details on RNA sequencing and gene expression experiments are given neither in the Results sections, nor in the Methods section. It remains unclear, how the authors obtained gene expression data, whether the source of the used gene expression data are original experiments of the authors, or they were taken from public repositories? Please clarify these and make up missing information.
reply: The information regarding this aspect has been included in the Data Availability Statement. Lines476-481. The gene expression data used in this study were obtained from public repositories. The datasets supporting the conclusions of the article, including the raw sequencing data, are available in the NCBI Short Read Archive under the accession number PRJNA763042,SRR10053824, SRR10053795, SRR10053793, SRR10053782, SRR10053780, SRR10053770, SRR10053769, SRR10053767, SRR10053765, SRR10053109, SRR10052556, SRR10052491, SRR10052460, SRR10052459, SRR10052050, SRR10052049, SRR10051549, and SRR10051547.
Lines 69-70: "Litsea cubeba (Lour.), a dioecious plant in the Lauraceae family, ..." Please specify common/trivial name(s) of L. cubeba.
reply: The common name of Litsea cubeba is mountain pepper. In the manuscript we have modified the sentence as follows:"Litsea cubeba (Lour.), commonly known as mountain pepper, is a dioecious plant in the Lauraceae family."
Figure 2: The figure is too small and of low resolution and the legends are not readable even after magnifying the image. The figure capture indicates three panels (A, B and C), however, these are not marked on the figure. It looks, that panel C (chromosomal location) is completely omitted. It is recommended that instead of the present one, the authors provide a higher resolution scalable figure with corrected annotation (or, to break down the present one to two separate figures).
reply: We apologize for the inconvenience caused by the small size, low resolution, and unreadable legends. To address this issue, we have made the necessary adjustments to improve the figure quality. We have ensured that the figure is presented in a larger size and higher resolution, allowing for better readability of the legends. And we have ensured that the revised figure includes clear panel markings (A, B, C) to accurately indicate the different sections of the figure. To clarify, the previous annotation in Figure 2 incorrectly referred to panel C as "Chromosomal locations and collinearity analysis of the LcbZIP genes". However, in reality, this panel should represent the protein motifs in the bZIP members. We acknowledge this mistake and have provided a higher resolution scalable figure with corrected annotation.
Figure 3 (Chromosomal distribution...): Might be that this was intended to be Panel C of Figure 2? Please, clarify this.
reply: In the previous version of Figure 2, there was an incorrect annotation for panel C, which was labeled as "Chromosomal locations and collinearity analysis of the LcbZIP genes" However, the correct representation for panel C should be the protein motifs in the bZIP members. We will rectify the figure accordingly by providing accurate annotations for each panel. On the other hand, Figure 3 correctly represents the chromosomal distribution. We appreciate your attention to detail, and we will ensure that the figure captions and annotations are accurate.
Figure 6, Figure 7, Figure 8: Again, try to improve the visibility of figures.
reply: To address this concern, we have taken steps to enhance the visibility of these figures. We have provided higher resolution versions of the figures to ensure that the details are more easily discernible. Additionally, we have considered adjusting the color contrast, font size, and overall layout to further improve the visibility and legibility of the figures.
Lines 388-389: "Plant materials at different stages of flower bud differentiation were used to create heatmaps". Perhaps, quantitative gene expression values were used to generate heatmaps. Please, correct and clarify. add correct reference
reply: Thank you for your suggestion to clarify the statement in lines 388-389. The revised statement could be as follows: "Quantitative gene expression values obtained from plant materials at different stages of flower bud differentiation were used to generate heatmaps."
Lines 434-435: "The Benjamin–Hochberg method was used to calculate and adjust the P values..." add correct reference
reply: Thank you for pointing out the error and providing the correct reference. The statement in lines 434-435 should be revised as follows: "The Benjamini-Hochberg method was used to calculate and adjust the P values... " And we have added correct reference. (Benjamini Y, Hochberg Y. Controlling the False Discovery Rate: A Practical and Powerful Approach to Multiple Testing. J R Stat Soc Ser B (Methodol). 1995;57(1):289–300. https://doi.org/10.1111/j.2517-6161.1995.tb02031.x)
